# Short-Arc Association and Orbit Determination for New GEO Objects with Space-Based Optical Surveillance

**Jian Huang** [1] , **Xiangxu Lei** [2,*] , **Guangyu Zhao** [3] , **Lei Liu** [3], **Zhenwei Li** [4], **Hao Luo** [5] and **Jizhang Sang** [3]

1 Beijing Institute of Tracking and Telecommunications Technology, Beijing 100094, China; huangjian@nudt.edu.cn
2 School of Civil and Architecture Engineering, Shandong University of Technology, Zibo 255000, China
3 School of Geodesy and Geomatics, Wuhan University, Wuhan 430079, China; gyzhao@whu.edu.cn (G.Z.); 20123016103302@whu.edu.cn (L.L.); zhsang@sgg.whu.edu.cn (J.S.)
4 Changchun Observatory of National Astronomical Observatory, Chinese Academy of Sciences, Changchun 130117, China; lizw@cho.ac.cn
5 Shanghai Astronomical Observatory, Chinese Academy of Sciences, Shanghai 200030, China; luoh@shao.ac.cn
* Correspondence: xxlei@whu.edu.cn; Tel.: +86-133-9606-4887

**Abstract:** For Geosynchronous Earth Orbit (GEO) objects, space-based optical surveillance has advantages over regional ground surveillance in terms of both the timeliness and space coverage. However, space-based optical surveillance may only collect sparse and short orbit arcs, and thus make the autonomous arc association and orbit determination a challenge for new GEO objects without a priori orbit information. In this paper, a three-step approach tackling these two critical problems is proposed. First, under the near-circular orbit assumption, a multi-point optimal initial orbit determination (IOD) method is developed to improve the IOD convergence rate and the accuracy of the IOD solution with angles-only observations over a short arc. Second, the Lambert equation is applied to associate two independent short arcs in an attempt to improve accuracy of the single-arc IOD semi-major axis (SMA) with the use of virtual ranges between the optical sensor and GEO object. The key idea in the second step is to generate accurate ranges at observation epochs, which, along with the real angle data, are then used to achieve much improved SMA accuracy. The third step is basically the repeated application of the second step to three or more arcs. The high success rate of arc associations and accurate orbit determination using the proposed approach are demonstrated with simulated space-based angle data over short arcs, each being only 3 min. The results show that the proposed approach is able to determine the orbit of a new GEO at a three-dimensional accuracy of about 15 km from about 10 arcs, each having a length of about 3 min, thus achieving reliable cataloguing of uncatalogued GEO objects. The IOD and two-arc association methods are also tested with the real ground-based observations for both GEO and LEO objects of near-circular orbits, further validating the effectiveness of the proposed methods.

**Keywords:** GEO objects; space-based optical surveillance; orbit determination with short-arc angle data; arc association; autonomous cataloguing; geosynchronous orbit



## 1. Introduction

It is fundamental for the safe use of near-Earth space to have the capability of routine, full space surveillance of space debris. Countries with strong space interests have established "national teams" for space surveillance to undertake monitoring, reconnaissance, and cataloguing of space objects [1]. Sensors on a Geosynchronous Earth Orbit (GEO) satellite can maintain stable ground pointing within a wide field-of-view (FOV). Benefiting from this property, GEO satellites are widely used in communications, reconnaissance, weather predication, defense applications, scientific applications, and so on. This has resulted in the deployment of more and more GEO satellites, and GEO orbit resources becoming more important strategically. Therefore, it is of great significance to fully catalogue GEO objects,

determine their orbit positions, prevent possible collisions of GEO objects, and analyze their orbital behaviors.

Ground-based optical telescopes have been primary facilities for detecting GEO objects, such as GEODSS [2], JAXA/IAT [3], AIUB ZIMLAT [4], Falcon [5], OWL-Net [6], FocusGEO [7], SSON [8,9], AGO70 [10], APOSOS [11], and so on. However, they are unable to detect and monitor GEO objects outside their effective FOV, and cataloguing the GEO objects over the full GEO region requires a global ground network, which may be unachievable for some countries. On the other hand, an optical surveillance satellite on a purposely designed low-altitude orbit may be able to survey the full GEO region. A surveillance satellite on a sun-synchronous orbit or a small-inclination orbit may also effectively suppress the effects of skylight and ground-reflected light to obtain an enhanced detection capability [12,13].

For uncatalogued GEO objects detected by space-based optical surveillance sensors, the most critical steps in their autonomous initial cataloguing are the arc association and orbit determination using the very first few arcs. A general procedure for the autonomous cataloguing of a new object is as follows. First, the identification of whether a detected object is a catalogued or uncatalogued object is made from the use of angle data over a short arc. For an uncatalogued object, the initial orbit determination (IOD) is performed with the short-arc observations, followed by the association of two independent arcs (determining whether the two arcs are from the same object), and finally, orbit determination using data from two or more arcs. For a catalogued object, its orbit can be updated with newly collected data together with earlier data. Clearly, it is essential to have high arc association correctness and accurate orbit determination solutions, since they are the basis for new object cataloguing, and the detection and identification of unusual orbit behaviors.

At the first step in cataloguing a new object, an IOD solution must be obtained from short-arc (less than 1% of orbital period) or very-short-arc (VSA, only 1–2 min for a GEO object or 10–20 s for an LEO object) angles. In fact, IOD results are the very base of the arc association in most cases [14]. For the IOD computation, there are several methods proposed by researchers. The traditional angles-only IOD methods (such as Gauss's method, double-r method, Laplace's method [15], and Gooding method [16]) applied to the VSA angles would probably fail due to the high observation noise and the short arc duration [17]. Several new methods have been proposed to tackle the VSA angles-only IOD problem. The method based on the concept of the Admission Region (AR) [14] provides a physics-based region of the range/range-rate space that produces Earth-bound orbit solutions. Further, DeMars et al. developed a method that employs a probabilistic interpretation of the AR and approximates the AR by a Gaussian mixture to obtain an IOD solution [18]. Gim and Alfriend proposed a geometric method to obtain the state transition matrix for the relative orbit motion that includes the effects of the reference eccentricity and the differential gravitational perturbations [19]. The result is useful for computing the primary gravitational perturbation that results from the gravity term $J_2$. DeMars et al. discussed a method for generating candidate hypotheses with angle-rate data for use in the multiple hypothesis filter [20]. Due to the lack of information on the range and range rate, the IOD with angular observations is of limited precision. Choi et al. derived the range information with two-site optical observations and the determined ranges have a high accuracy [21]. Weisman et al. presented an approach to estimate the uncertainty or probability density function (PDF) associated with the state vector for space objects in LEO [22]. The method is used to initialize conventional non-linear filters, as well as to operate a Bayesian approach for orbit determination and object tracking. Maruskin et al. presented a new method by taking into account certain physical considerations, and the orbit can be mapped with high precision without an excessive computational burden [23]. The method reduces the orbit determination process to performing intersections of two-dimensional laminas in the plane. Sciré et al. showed that the batch estimators could be a useful tool to estimate the state of space debris at a certain time [24]. Tao et al. presented a more accurate IOD method [25], namely the Laplace-LS orbit determination method,

whose estimation variance is close to the Cramer–Rao Lower Bound (CRLB). It can be used when the observation arc is very short or the sensor has limited accuracy. Porfilio et al. reported a two-site optical observation campaign addressed to the orbit determination of objects in GEO without a priori information carried out by University of Rome "La Sapienza" (GAUSS) [26].

The previous researchers tried to solve the IOD problem in theoretical ways to obtain analytical solutions, while Sang et al. proposed a range-search IOD method, which assumes ranges at two chosen epochs and then solves the Lambert problem, where a residual control process is employed to control the quality of the IOD solutions [27]. Processing both real ground-based and simulated space-based VSA angle data shows that the method has an IOD success rate over 90%. However, the errors of the solutions are usually large, such that a solution from angles data over a single short arc is basically useless if it cannot be associated with another arc. To get the errors present in initial orbit elements of the space debris with angle observations, the limitations of the orbit determination methods must be understood well [28]. Using observations obtained by Lockheed Martin's Space Object Tracking (SPOT) facility, Stoker et al. analyze the effectiveness of angles-only orbit determination methods with limited observations, and the error in each IOD method shows a strong correlation with the amount of observation arcs [28].

When the angle data of a GEO object is collected by a sensor on a low-altitude satellite platform, the autonomous arc association and orbit determination are generally more difficult. This is due to the high orbiting velocity of the satellite platform: the lengths of observed arcs for a GEO object are usually very short with regard to the orbit period of the observed object. Typically, an arc of about 3–5 min for a GEO object would be observed by an optical sensor of 2 degrees FOV flying on a nonsynchronous orbit at an altitude of 600 km. As a result, high IOD convergence rate and accuracy of the IOD cannot be guaranteed [29]. When the observed arcs or IOD tracks cannot be associated to any object in the catalogue, they are usually regarded as uncorrelated tracks (UCTs). It is essential to associate two or more UCTs to the same uncatalogued object, such that all associated UCTs are processed altogether to obtain accurate orbit elements of the object. Generally, if three or four UCTs are correlated correctly to an uncatalogued object, the orbit estimated using the multiple arcs would be accurate enough to catalogue this object [30]. To some extent, the methods correlating the UCTs [30,31] have various limitations. Most existing methods [23,32] require error information of IOD solutions to perform the track association, which is usually unavailable or unreliable. The effectiveness of the geometrical approach [31] depends on the accuracy of IOD solutions. All these factors make the autonomous association and initial orbit cataloguing a real challenge.

This paper proposes a three-step approach to achieve high-performance autonomous cataloging of newly detected GEO objects using the space-based short-arc angular data. First, considering the near-circularity of orbits of most GEO objects, a multi-point IOD optimization method is developed to determine a preliminary set of orbit elements from the angle data of a single short arc. Second, the Lambert equation is applied to associate two independent short arcs in an attempt to improve the accuracy of the single-arc IOD semi-major axis (SMA) with the use of virtual ranges between the optical sensor and GEO object. The key idea in the second step is to generate accurate ranges at observation epochs, which, along with the real angle data, are then used to achieve much improved SMA accuracy. As a result, the correlation of two arcs is determined, and more accurate orbit elements from the use of two arcs are estimated. In the third step, a third arc or more arcs are combined to further improve the accuracy of the orbit elements estimated in the second step, and the object can then be subsequently catalogued.

In the following, the three-step approach is presented in Section 2, and the results in Section 3. Section 4 concludes this paper.

## 2. Computation Scheme

### 2.1. A Multipoint Optimal Angles-Only IOD Method for Near-Circular GEO Orbit

The general procedure (Figure 1) to associate two short arcs involves, first, the IOD of each arc, and then the determination of the correlation of the two IOD orbit element sets. That is, the orbit elements are the basis for autonomous association of orbit arcs, and high-precision orbit elements would be useful for reliable association and subsequent orbit improvement. In addition, the accurate IOD may be essential for fast object identification, maneuver detection, short-term orbit prediction, and so on. For GEO objects, space-based optical surveillance may only generate an arc as short as a few minutes.

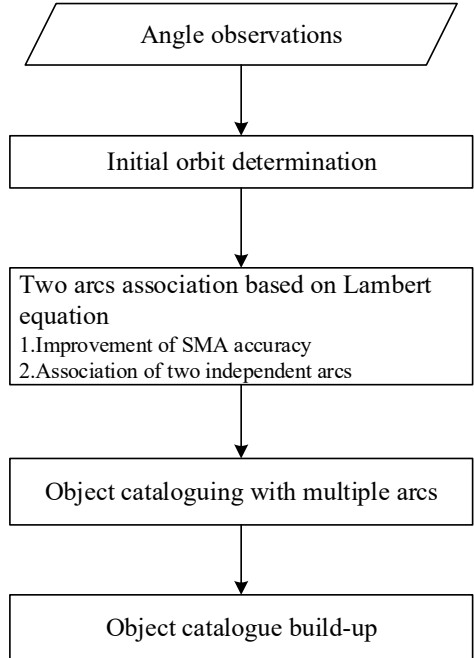

**Figure 1.** The procedure of the method in this paper.

Generally, the IOD would need an arc length longer than 1% of the orbital period (that is, about 15 min for GEO objects), and then the improved-Laplace [33], Gauss [15], or Gooding [16] methods are likely used to generate stable IOD solutions. Otherwise, ill-conditioned equations in these methods make the IOD difficult to converge [34,35]. The use of the range-search-based IOD method [27] may have the problems of expansive search time and solution optimization.

#### 2.1.1. IOD with Angular Observations at Two Arbitrary Epochs

In order to improve the convergence rate of the traditional IOD methods and the solution accuracy, this paper uses a characteristic of GEO orbits as prior information in the determination of the IOD elements. That is, the GEO orbit eccentricity is usually very small, so that it can be assumed as a circular orbit in the IOD. With this assumption, and given angular observations at two epochs, an iterative search of the semi-major axis (SMA), $a$, can be performed, in which an objective function is used to constrain the angular velocity of orbital motion [36]. The objective function is:

$$\Delta n(a) = n_1(a) - n_2(a) = 0 \tag{1}$$

where,

$$n_1(a) = \sqrt{\frac{\mu}{a^3}} \quad n_2(a) = \arccos\left(\frac{\vec{r}_1 \cdot \vec{r}_2}{a^2}\right) \frac{1}{\Delta t}\left[1 + \frac{3J_2}{4a^2}\left(6 - 8\,\sin^2 i\right)\right]$$

In Equation (1), $\mu$ is the Earth's gravitational constant; $J_2$ the second order term of the Earth's gravitational expansion; $\vec{r}_1$ and $\vec{r}_2$ the geocentric position vectors at two observation epochs, respectively; $\Delta t$ the time interval between the two epochs; and $i$ the inclination of the orbit plane.

Equation (1) holds or nearly holds if the SMA is close to its truth. However, the SMA is unknown and to be determined. Without the range information, the angles at two epochs are insufficient to solve the SMA. With the zero-eccentricity assumption, if the SMA, which is the same as the radius of the orbit, is given, then $\vec{r}_1$ and $\vec{r}_2$ (and thus the other orbit elements) can then be computed, because the ranges between the sensor and object at the observation epochs can be determined from the assumed SMA, as well as the known position of the sensor and angles from the sensor to the object at the epochs, as shown in Figure 2. According to Figure 2, the formulae to compute the range between the sensor and object are as follows:

$$\rho = \frac{a}{\sin\theta}\sin\gamma \tag{2}$$

where

$$\theta = \pi - \arccos\left(\vec{u}\cdot\frac{\vec{r}_s}{r_s}\right)$$
$$\vec{u} = \begin{pmatrix} \cos\delta\cos\alpha \\ \cos\delta\sin\alpha \\ \sin\delta \end{pmatrix}$$
$$\gamma = \pi - \theta - \beta$$
$$\beta = \arcsin\left(\frac{r_s}{a}\sin\theta\right)$$
$$r_s = \|\vec{r}_s\|_2$$

where $\alpha, \delta$ are the right ascension (RA) and declination (DEC) of the object with respect to the sensor, $\vec{r}_s$ is the position vector of the sensor, and $\|\bullet\|_2$ represents the second order norm of a vector.

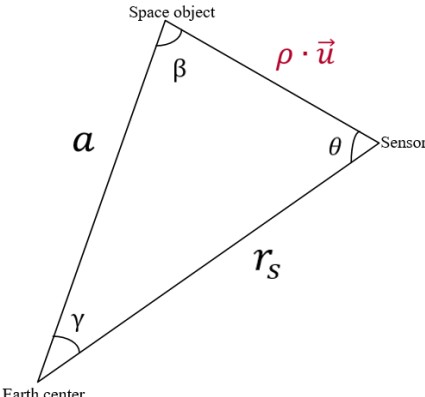

**Figure 2.** Geometry to compute range between sensor and object.

An extensive search on the SMA of the object orbit would find the SMA to make Equation (1) hold. For a GEO object, the SMA is about 42,000 km. This provides a good base for the SMA search. This paper uses a segmentation method to try the SMA and evaluate Equation (1). The segmentation method is as follows.

(1) Assume the SMA is in the range $[a_1, a_n]$, and divide this range into N sub-ranges each having a length $\Delta a = (a_n - a_1)/N$. The $ith$ sub-range is then $[a_i, a_i + \Delta a]$, $a_i = a_1 + (i-1)\Delta a$. For an object in the GEO orbit region, $a_1$, $a_n$, and $\Delta a$ may be set to 40,000 km, 44,000 km, and 50 km, respectively.

(2) For each sub-range, compute the objective function values at its lower and upper boundary SMAs, $a_l$ and $a_u$, respectively, which results in $\Delta n(a_l)$ and $\Delta n(a_u)$.

(3)   If the two function values have the same sign, then the true SMA is not in this sub-range; return to Step 2 to assess the next sub-range;

(4)   Otherwise, if the two function values have the opposite sign, this sub-range is divided into two segments of equal length; then, return to Step 2 to assess the new sub-ranges.

(5)   Step 4 is terminated when the function values at the lower and upper boundaries of a SMA sub-range are close to each other within the preset threshold, and the mean of the two boundary values of the sub-range is an estimate of the SMA of the object orbit.

The above procedure of SMA trial and evaluation of the objective function has a very high convergence rate. However, influenced by the errors of angular observations and the short length of the time interval between the two epochs, the uncertainty of the estimated orbit elements is usually large, and it is reasonable to ask if the uncertainty could be reduced through the use of angle data at more epochs.

### 2.1.2. Quality Assessment of IOD Orbit Elements Using Observation Sequence

An optically surveyed orbit arc will have a sequence of multiple data points. Using any two points could generate an IOD orbit solution, and thus, a number of orbit solutions could be obtained through the combination of two-point observations. An optimal set of IOD elements may be determined with an optimization process to all these solutions. Assume there are angular observations in the form of the right ascensions (RAs) $\{\alpha_1, \alpha_2, \cdots, \alpha_m\}$ and the declinations (DECs) $\{\delta_1, \delta_2, \cdots, \delta_m\}$ in an inertial coordinate system at m epochs $\{t_1, t_2, \cdots, t_m\}$. Repeated random choosing of two epochs results in an assembly, $\{\Theta_1, \Theta_2, \cdots, \Theta_M\}$, where $\Theta_i = \{t_{i_1}, \alpha_{i_1}, \delta_{i_1}, t_{i_2}, \alpha_{i_2}, \delta_{i_2}\}$, $(i_1, i_2) \in \{1, 2, \cdots, m\}$, and $i_1 \neq i_2$, where M is the number of the members in the assembly. Suppose $\sigma_i(t_0)$ is the IOD orbit element set at $t_0$ solved from the i-th member of the assembly. Affected by the measurement errors, the M sets of orbit elements have different levels of uncertainty. This means that some of them are closer to the truth values than others. Therefore, it is necessary to determine an optimal set of orbit elements from the M sets. But first, the quality of an IOD element set has to be assessed to reject those poor-quality IOD element sets. The IOD element quality can be assessed based on the residuals of the angle data. Given an orbit element set, the RAs and DECs at the m observation epochs can be computed, resulting in $\{\hat{\alpha}_1, \hat{\alpha}_2, \cdots, \hat{\alpha}_m\}$ and $\{\hat{\delta}_1, \hat{\delta}_2, \cdots, \hat{\delta}_m\}$. The residuals are then:

$$\left\{ \begin{array}{c} \Delta\alpha_j = (\hat{\alpha}_j - \alpha_j)\cos\delta_j \\ \Delta\delta_j = \hat{\delta}_j - \delta_j \end{array} \right., \quad j = 1, 2, \ldots, m \tag{3}$$

It is then easy to compute the RMS (root mean square) values, $\alpha_{rmse}$ and $\delta_{rmse}$, of the RA and DEC residuals, respectively. Further, these two residual sequences are fitted into two linear functions:

$$\left\{ \begin{array}{c} \Delta\alpha_j \approx c_0 + c_1(t_n - t_0) \\ \Delta\delta_j \approx b_0 + b_1(t_n - t_0) \end{array} \right., \quad j = 1, 2, \ldots, m \tag{4}$$

The least-squares estimation is made to determine the coefficients $c_0$, $c_1$, $b_0$, and $b_1$ in Equation (4), where $c_0$ and $b_0$ represent the systematic deviations of the computed RAs and DECs from the observations, respectively, and $c_1$ and $b_1$ are the deviation rates (DRs) of the computed RAs and DECs from the observations, respectively.

Now, we may assess the quality of the estimated IOD orbit elements using the RMS and DR values, since, the smaller these values, the better the agreement between the IOD orbit and the observations. If conditions in Equation (5) below are met, the corresponding IOD orbit element set is regarded as a quality IOD solution:

$$\left\{ \begin{array}{c} \alpha_{rmse} \leq \text{RMS}_{\text{IOD,TH}}, \text{ and } \delta_{rmse} \leq \text{RMS}_{\text{IOD,TH}}, \text{ and} \\ |c_1| \leq \text{DR}_{\text{IOD,TH}}, \text{ and } |b_1| \leq \text{DR}_{\text{IOD,TH}} \end{array} \right. \tag{5}$$

where $RMS_{IOD,TH}$ is the RMS threshold, and $DR_{IOD,TH}$ the DR threshold. The thresholds for the RMSs of the RA and DEC residuals are the same, as well as the thresholds for DRs of the RA and DEC residuals, since the accuracy of the RA and DEC observations is usually the same.

### 2.1.3. Determination of an Optimal Orbit Element Set

The resultant quality IOD orbit element sets are further processed to generate an optimal IOD solution. In this process, only the IOD solutions with small DR values are used. First, the sum of $|a_1|$ and $|b_1|$ of each of the quality element sets is obtained, and the sums are ordered ascendingly. Then, the first few—for example, the first one-tenth—of the ordered sets are chosen to go through the averaging process. Assume the first $P$ element sets are chosen, and set a reference epoch, say $t_0$. Let $\sigma_i$ be a chosen orbit element set. Then, it can be used to compute the position and velocity vectors of the object at $t_0$ using the two-body orbit theory [37] as:

$$\begin{cases} \vec{r}_{i,t_0} = R(\sigma_i, t_0) \\ \vec{v}_{i,t_0} = V(\sigma_i, t_0) \end{cases} , \quad i = 1, 2, \ldots, P \tag{6}$$

where $R$ and $V$ are the formulas for computing the position vector and velocity vector from the orbit elements, respectively. Therefore, the mean of the $P$ position vectors, $\vec{r}_{mean}(t_0)$, and the mean of the $P$ velocity vectors, $\vec{v}_{mean}(t_0)$, can be obtained, and they are converted to the Kepler orbit elements, $\sigma_{mean}(t_0)$, in the following form:

$$\sigma_{mean}(t_0) = f\left(\vec{r}_{mean}, \vec{v}_{mean}, t_0\right) \tag{7}$$

We have now obtained an optimal IOD solution, $\sigma_{mean}(t_0)$, from angle data over a short arc.

### 2.2. Association of Two Arcs Based on Lambert Equation

The short-arc IOD accuracy is limited due to the short observation duration, and it is extremely difficult to improve the accuracy if no further information is available to use. One option is to combine two arcs together in the orbit determination. If two arcs are from the same object, the orbit improvement is certainly expectable because of the extended length between two arcs. Before that, however, the correlation between the two arcs has to be determined, indicating that the arc association is a critical step to the autonomous cataloguing of new space objects.

Usually, whether two arcs are correlated is determined by the hypothesis test on the two IOD orbit element sets, in which the orbital mechanics constraints on the IOD orbit elements are applied [30,38–41]. In case there are redundant observations, a self-consistency can be performed, such as in the inter-screen correlation of dual-screen radar [42], which has multiple position vectors for orbit determination and redundant velocity information to verify the orbit determination results. In this manner, a very high true positive rate of 97% is achieved when performing two-arc association [42].

However, the space-based optical surveillance will most likely collect only sparse orbit arcs for GEO objects. In this case, there is no redundant information, and more critically, the range information important for the orbit determination is unavailable. Consequently, the autonomous arc association would be much more difficult [32,43].

In this paper, a method of two-arc association based on the use of Lambert equation is proposed, in which the observation change trends are evaluated to determine the correlation of two arcs.

### 2.2.1. Improvement of SMA Accuracy by Application of the Lambert Equation to Two Arcs

The first step in the two-arc association is to compare the SMAs of the two IOD orbits and normal vectors of the two IOD orbit planes. When the differences in the SMAs and the

angle between the two normal vectors are less than the preset thresholds, the two arcs will be further assessed for their correlation.

Before proceeding to the details of the method, which extensively applies the Lambert equation, it is noted that the Lambert equation holds only for the two-body orbit; therefore, it is necessary to justify the applicability of the Lambert equation to two position vectors of a GEO object apart by a few days. Here, only the secular perturbation due to dominant $J_2$ term is considered.

The $J_2$-induced secular rates of the SMA, eccentricity, and inclination of an Earth-orbiting object's orbit are zero, and those of the right ascension of ascending node (RAAN), perigee argument, and mean anomaly are [37]:

$$\dot{\Omega} = -\frac{3}{2}\frac{J_2 R_E^2 n}{a^2(1-e^2)^2}\cos i \quad \text{the rate of the RAAN} \tag{8}$$

$$\dot{\omega} = \frac{3}{4}\frac{J_2 R_E^2 n}{a^2(1-e^2)^2}\left(4 - 5\sin^2 i\right) \text{the rate of the perigee argument} \tag{9}$$

$$\dot{M} = \frac{3}{4}\frac{J_2 R_E^2 n}{a^2(1-e^2)^{3/2}}\left(2 - 3\sin^2 i\right)\text{the rate of the mean anomaly} \tag{10}$$

where, $n = \sqrt{\frac{\mu}{a^3}}$ is the mean motion, $R_E = 6,378,137$ m the Earth radius, and $e$ the orbit eccentricity. For the GEO orbit, we can assume $a = 36,000$ km $+ R_E$, $e = 0$, $i = 0$, $J_2 = 1.08263 \times 10^{-3}$, and $\mu = 3.986 \times 10^5 \text{km}^3/\text{s}^2$. This results in $\dot{\Omega} = -2.7 \times 10^{-9}/\text{s}$, $\dot{\omega} = 5.4 \times 10^{-9}/\text{s}$, $\dot{M} = 2.7 \times 10^{-9}/\text{s}$. For the time interval of 3 days, the secular variations of the RAAN, the perigee argument, and the mean anomaly caused by $J_2$ are about 140″, 280″, and 140″, respectively.

It is noted that the main objective of applying the Lambert equation to two positions from two arcs is to determine a set of orbit elements with an accuracy sufficient to determine the association of the two arcs. Although the secular perturbation induced by $J_2$ over 3 days causes the real orbit to deviate from the two-body orbit, the deviation in the form of the above secular variations in the RAAN, the perigee argument, and the mean anomaly may still make the Lambert equation applicable to two arcs, even when separated by 3 days, with a loss of accuracy in the estimated elements as the cost.

Simulation experiments are made to verify the applicability of the Lambert equation to two position vectors of a GEO object. First, 100 two-position pairs are generated for 100 GEO objects using the TLEs of the objects. That is, one pair is for one object. The two positions in a pair are processed with the Lambert equation, and the solved SMA is compared to the SMA in the TLE of the object. The results show that, when the interval between two positions is longer than 12 h but less than 72 h, 59.60% of the SMA differences are less than 3 km, and 63.87% of them are less than 5 km. When the time interval is longer, the Lambert method induces a larger error because the actual orbit deviates more seriously from the two-body orbit. That is, the use of the Lambert equation in the GEO orbit is better limited to two positions separated by less than 72 h. In the following, two arcs to be associated are required to be less than 72 h apart.

Now, suppose $\sigma_{mean}(t_1)$ is the IOD orbit element set obtained from the first arc at $t_1$, the position vector $\vec{r}_1$ at the epoch of $t_1$ is computed by Equation (6). In the same way, the position vector $\vec{r}_2$ at $t_2$ with $\sigma_{mean}(t_2)$ of the second arc is computed. The Lambert equation in the two-body problem is expressed as [37,44]:

$$t_2 - t_1 = \left(\frac{a^3}{\mu}\right)^{\frac{1}{2}}[(\psi - \sin\psi) - (\varepsilon - \sin\varepsilon)] \tag{11}$$

Given $r_1 = \|\vec{r}_1\|_2$, $r_2 = \|\vec{r}_2\|_2$, and $c = \|\vec{r}_2 - \vec{r}_1\|_2$, $\psi$ and $\varepsilon$ are then computed by

$$\begin{cases} \cos\psi = 1 - \frac{r_1+r_2+c}{2a} \\ \cos\varepsilon = 1 - \frac{r_1+r_2-c}{2a} \end{cases} \tag{12}$$

The SMA, $a$, can now be solved from Equations (11) and (12) iteratively, with the initial value of $a$ taken from the IOD elements of the first arc or second arc. When the time interval $t_2 - t_1$ is more than 12 h, the accuracy of the solved $a$ could be at the kilometer level. In addition, the other orbit elements can be computed from the two position vectors, and the full set of the orbit elements is denoted as $\sigma_{Lambert}(t_0)$.

2.2.2. Association of Two Independent Arcs

The use of Equations (11) and (12) to solve $a$ has assumed that the two position vectors $\vec{r}_1$ and $\vec{r}_2$ are of the same orbit, that is, the two arcs are from the same object. In reality, the two arcs could be from two different objects, and if this is the case, the solved $a$ is of no meaning. This demands the determination whether the two arcs are from the same object. We propose to apply the least-squares method to determine the orbit elements from using all angle data of the two arcs; then, the correlation of the two arcs is judged by the analysis of the angles residuals.

When only angle data of two short arcs is available, there is no guarantee that the least-squares orbit determination will be converged or the accuracy of the determined orbit elements will be satisfactory. This is due to the weak geometrical constraints on the orbit provided by the sparse angle data [29,45]. The geometrical strength could be enhanced if additional information is used. For the GEO objects this paper is concerned with, the characteristics that the orbit eccentricity of a GEO object is nearly zero and the accuracy of the semi-major axis solved from two separated arcs (of the same object) is relatively high are explored to generate virtual ranges between the optical sensor and the object with Equation (2).

With the computed virtual range $\rho$, the three-dimensional measurements $(\rho, \alpha, \delta)$ at the observation epochs of the two arcs can now be processed in the least-squares orbit determination process, in which $\sigma_{Lambert}(t_0)$ provides the initial values. The general observation equations for the measurements $(\rho, \alpha, \delta)$ at $t$ are expressed as [35]:

$$[\rho(t), \alpha(t), \delta(t)] = H(\sigma_{Lambert}(t_0), t - t_0) + V \tag{13}$$

where $H$ are the functions to compute $(\rho, \alpha, \delta)$ at $t$ from orbit elements $\sigma_{Lambert}(t_0)$, and $V$ the measurement errors.

The use of these functions involves orbit propagation with $\sigma(t_0)$ from $t_0$ to $t$, during which perturbation forces can be taken into consideration. In this study, only the Earth gravity of an order/degree up to 5 and solar/lunar gravity are considered. More details can be found in Liu and Tang [35].

It should be noted that the above least-squares orbit determination is generally able to generate accurate orbit elements if the two arcs are of the same orbit, largely because of the use of the virtual ranges.

To confirm the correlation of the two arcs, the angles residuals of each arc are again fitted using Equation (4), and the coefficients $c_0(t_1), c_1(t_1), b_0(t_1), b_1(t_1)$ for the first arc and $c_0(t_2), c_1(t_2), b_0(t_2), b_1(t_2)$ for the second arc are obtained. Similar to the IOD quality assessment where the residual deviation rates are assessed, the quality of the least-squares orbit determination using the angle data of the two arcs is assessed with the following criteria:

$$\begin{cases} c_1(t_1) \le DR_{LS,TH}, \text{ and } b_1(t_1) \le DR_{LS,TH}, \text{ and} \\ \quad c_1(t_2) \le DR_{LS,TH}, \text{ and } b_1(t_2) \le DR_{LS,TH} \end{cases} \tag{14}$$

where $DR_{LS,TH}$ is the threshold for the DR of the residuals.

When Equation (14) holds, the two arcs are very likely from the same object, and their correlation is declared.

### 2.3. Object Cataloguing with Multiple Arcs

The two-arc association process lays the foundation for robust object cataloguing which requires at least three arcs. Assume that two arcs have been correctly associated and the method in Section 2.2.2 has generated a least-squares orbit solution $\sigma_{LS,\,2}(t_0)$, and a third arc has been associated to either of the two arcs. It is natural to repeat the procedure presented in Section 2.2.2 for the three arcs, in which $\sigma_{LS,\,2}(t_0)$ is used as the initial values for the least-squares orbit determination, resulting in the orbit solution from the three arcs, $\sigma_{LS,\,3}(t_0)$. If the quality assessment indicates the success of the least-squares orbit determination, the accuracy of $\sigma_{LS,\,3}(t_0)$ would be higher than that of $\sigma_{LS,\,2}(t_0)$. This procedure can be repeated for the fourth arc, fifth arc, and so on. The final result is that the stable orbit accuracy is achieved when more and more arcs are processed together, and eventually the object is successfully catalogued.

### 2.4. Algorithm Implementation

Given a pool of arcs of angle observations, the proposed procedure can be implemented to associate any two arcs and determine a set of accurate orbit elements from a number of arcs in the following steps:

1. Apply the IOD method in Section 2.1 to each single arc to obtain a set of IOD elements for the arc.
2. Given two arcs, denoted as Arc1 and Arc2, if the two arcs are apart by less than a preset time interval threshold (e.g., 3 days), and the difference in the SMAs of the two arcs and the angle between the two normal vectors of the two IOD orbit planes are less than the preset thresholds, the two arcs will be further assessed for their correlation using Steps 3 and 4.
3. Apply the Lambert problem method in Section 2.2.1 to the two arcs to determine a set of orbit elements, denoted as $\sigma_{LP,2}$.
4. Apply the method in Section 2.2.2 to determine a new set of orbit elements from the use of all data of the two arcs, in which $\sigma_{LP,2}$ are used as the initial values in the least-squares orbit determination. If the quality test in Equation (14) passes, Arc1 and Arc2 are very likely from the same object; their association is declared, and the resulting orbit elements are denoted $\sigma_{LS,2}$.
5. For another arc, denoted as Arc3, if it is associated to either Arc1 or Arc2, the three arcs are processed using the method in Section 2.3. If it is successful, they can be declared to be from the same object, and the determined orbit elements are more accurate than $\sigma_{LS,2}$.
6. Repeat Step 5 to process a fourth, fifth, ..., arc. When a new arc is included in the orbit determination, and the quality test passes, the new arc is successfully associated, and accurate orbit elements are determined from the use of data of all arcs.

## 3. Results

### 3.1. Angle Data and Threshold Settings

The developed algorithm is tested with both simulated data of GEO objects observed by a simulated space-based telescope and real data from three ground-based optical sensors.

The real data of GEO objects are observed by two ground-based optical sensors. The first is an electrical-optical telescope array (EA) at Changchun Observatory, and the second is the FocusGEO developed by Shanghai Astronomical Observatory (SAO). The Changchun GEO EA has four telescopes—each has an aperture of 28 cm, focus length of 32.4 cm, and a FOV of $6.5° \times 6.5°$ [46]. A total of 1542 arcs from the EA, collected over three days from 6 February 2021 to 8 February 2021, will be processed. These arcs are from 234 objects, and they are all longer than 30 s with at least 5 data points. The mean duration of the arcs is about 70.6 s with 19 data points. The measurement error is about 2.4″.

The SAO FocusGEO (left, Figure 3) is a new-generation telescope dedicated for GEO objects. A total of 1941 arcs, collected over 4 days from 20 October 2019 to 23 October 2019, are to be processed. These arcs are from 59 objects. The mean duration of the arcs is about 44.8 s with 8 data points. The measurement error is about 4.2". More information on the FocusGEO can be found in reference [7].

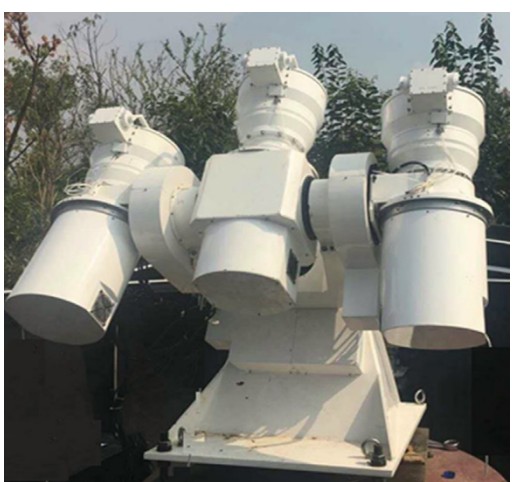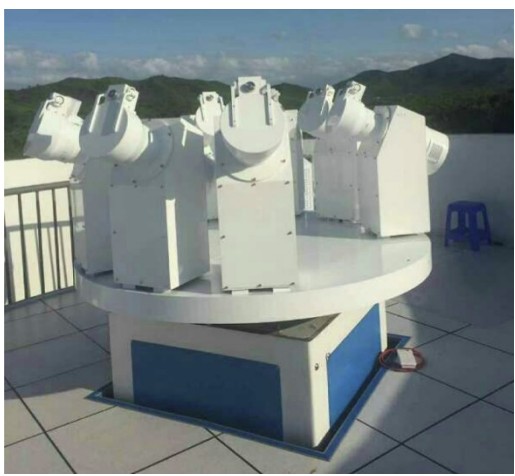

**Figure 3.** Physical pictures of FocusGEO [7] (**left**) and Changchun LEO EA (**right**).

The proposed IOD method is also tested with observations of LEO objects from another ground-based optical EA sensor at Changchun Observatory. The LEO EA (right, Figure 3) has eight small telescopes—each has an aperture of 15 cm and a FOV of $14° \times 14°$. A total of 10,621 arcs from the EA, collected over three days from 24 August 2017 to 26 August 2017, are processed, and 9017 of these arcs are correctly correlated with 1961 LEO objects of TLE data. About 95.38% of the correlated arcs are from objects at near-circular orbit (the eccentricity less than 0.05). The mean duration of the arcs is about 39.5 s with 22 data points. The measurement error is about 9.0".

The space-based simulation data is generated assuming that a telescope is onboard a satellite on a Sun-synchronous orbit (SSO) with an altitude of 660 km. The eccentricity of the SSO is nearly zero. The telescope is fixed to the satellite body and pointed towards the GEO ring at an inclination of $0°$, and the FOV of the camera is $3° \times 3°$. Assume that there are 691 GEO objects, and their orbits are known. With this setup, when the satellite orbits the Earth, the telescope FOV will cover a GEO orbit region when the satellite crosses the equator, objects in the GEO region will be surveyed, and the angle data in the form of the RA and DEC from the satellite to the objects are collected. In this simulation, 3235 arcs from 691 GEO objects over 10 days are collected, and the sampling interval of angle data is 3 s. Figure 4 shows a GEO arc observed from an LEO platform. The measurement noise is 10" for both the RA and DEC angles. To test the effectiveness of the proposed approach for cataloguing objects from short arcs, the arc length is set to 3 min.

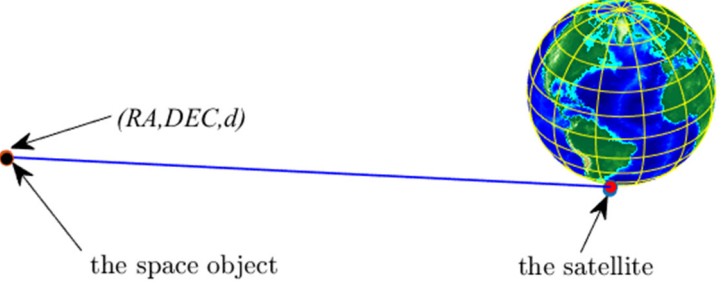

**Figure 4.** Illustration of an observed GEO arc from an LEO platform (the term "d" in the figure means the distance, and it is unknown for optical surveillance).

For the assumed measurement noise of 10″ in the simulated space-based data, the following thresholds are set as:

The criteria in Equation (5) for accepting IOD solution are:

$$\text{RMS}_{\text{IOD,TH}} = 200''$$
$$\text{DR}_{\text{IOD,TH}} = 5''\ /\text{minute}$$

The criterion in Equation (14) for declaring association of two arcs is:

$$\text{DR}_{\text{LS,TH}} = 5''\ /\text{minute}$$

It is noted that these thresholds are determined through extensive experiments with the assumed accuracy and arc length of the simulated space-based data. In the experiments, we have tried different thresholds, and found that the above thresholds are appropriate for the problem we face. For example, when $\text{DR}_{\text{LS,TH}}$ is increased or reduced by 10% from the used 5″/minute, the TP rate will change by less than 1.5%. It is also noted that the same thresholds are used in processing the real data.

### 3.2. IOD Experiments

The IOD computation is performed for the 3235 GEO orbit arcs of simulated angular data. The used IOD methods include the improved-Laplace method (i-Laplace method) [23], the range-search based IOD method (RS method) [27]), and the one proposed in this paper. Table 1 presents the statistics of the SMA errors, and the IOD success rates. The IOD is claimed a success if an optimal estimate of the SMA is found and the error of the estimated SMA is less than 1000 km. This threshold of 1000 km has an insignificant effect on the success rate of the proposed IOD method, since more than 93% of estimated SMAs have errors less than 200 km (the last row in Table 1). It is seen that the proposed method is superior to the other two angles-only IOD methods for 3 min GEO orbit arcs, in terms of the SMA errors and the success rate. The IOD success rate of 98.25% means that only 56 of 3235 arcs will not take part in the two-arc association.

**Table 1.** Performance of processing simulated GEO observations with some IOD methods.

| Method | SMA Errors | | | | Success Rate |
| --- | --- | --- | --- | --- | --- |
| | ≤20 km | ≤50 km | ≤100 km | ≤200 km | |
| i-Laplace method [33] | 12.38% | 26.97% | 43.44% | 61.14% | 85.98% |
| RS method [27] | 38.36% | 64.60% | 75.73% | 79.75% | 86.26% |
| Proposed method | 41.24% | 72.15% | 86.25% | 93.11% | 98.25% |

The main results from processing the real data with the proposed IOD method are presented in Table 2. The IOD success rates are 99.66% and 98.08%, respectively, for the data observed by Changchun GEO EA and SAO FocusGEO. However, the IOD accuracy for the SAO FocusGEO data is worse. This is propably due to the relatively shorter arc length of the SAO FocusGEO data. In terms of the success rate, the proposed IOD method has it above 98% for both the simulated space-based and real GEO data, validating the applicability of the proposed IOD method for processing angle data of single arcs of GEO objects. The success rate is 95.81% for data of LEO objects at near-circular orbits, demonstrating its strong applicability.

**Table 2.** Performance of processing ground-based observations with the proposed IOD method.

| Sensor | SMA Errors | | | | Success Rate |
| --- | --- | --- | --- | --- | --- |
| | ≤20 km | ≤50 km | ≤100 km | ≤200 km | |
| Changchun GEO EA | 34.64% | 74.93% | 90.23% | 98.79% | 99.66% |
| SAO FocusGEO | 15.27% | 34.78% | 59.75% | 85.99% | 98.08% |
| Changchun LEO EA | 48.39% | 87.40% | 96.45% | 100.00% | 95.81% |

### 3.3. Two-Arc Association

The proposed two-arc association method is based on the use of the Lambert equation and the least-squares estimation of orbit elements; thus, it is necessary to examine the effectiveness of the method when it is applied to two arcs separated by up to 3 days, in addition to the validation presented earlier, which was focused on the applicability of the Lambert equation. In this validation, two arcs from two different objects are also processed. The separation interval between two arcs is again limited to within 3 days. The SMA in the TLE is still used as the reference to estimate the error of the SMA determined from the use of the Lambert equation. For each of the cases of different separation times, the medians of about 1000 differences between the reference and estimated SMAs are given in Table 3.

**Table 3.** SMA differences of processing two arcs of simulated GEO observations.

| Two Arcs of | Separation Time (h) | SMA Difference (km) |
|---|---|---|
| The same object | (0, 12) | 1.27 |
| | (12, 24) | −1.40 |
| | (36, 48) | 1.35 |
| Different objects | (0, 12) | −2529.70 |
| | (12, 24) | −2793.73 |
| | (36, 48) | −2282.68 |

It is seen from Table 3 that when two arcs are of the same object, the SMA differences are less than 2 km. However, when two arcs are of two different objects, the SMA differences are thousands of km. Similarly, other orbit elements have much larger differences if two arcs are from two different objects. Therefore, if two arcs are from the same object, the orbit elements of the object will be estimated with a high accuracy, which will most likely make the parameters in Equation (11) less than the preset threshold. Otherwise, the parameters will be significantly larger than the threshold. These findings suggest that the parameters in Equation (11) are well suited to determining the orbit association of two arcs.

Applying the proposed method to determine orbit elements from real data of two arcs, we obtain the differences between estimated SMA and TLE SMA given in Table 4. For GEO objects, the differences are 18.7 km and 32.86 km, respectively, for Changchun GEO EA and SAO FocusGEO, when the time interval between two arcs of the same object is less than 12 h. The differences are less than 5 km when the interval is longer than 12 h. However, the differences are more than 118 km when two arcs are from different objects, which are significantly larger than those for two arcs of the same object.

**Table 4.** SMA differences of processing two arcs of real ground-based observations.

| | | SMA Difference (km) | | |
|---|---|---|---|---|
| Two Arcs of | Separation Time (h) | Changchun GEO EA | SAO FocusGEO | Changchun LEO EA |
| The same object | (0, 12) | 18.70 | 32.86 | 2.98 |
| | (12, 24) | 0.86 | 4.03 | 4.20 |
| | (36, 48) | 1.44 | −1.35 | 4.49 |
| Different objects | (0, 12) | −1144.99 | −1974.79 | −27.51 |
| | (12, 24) | 296.63 | −118.82 | −285.05 |
| | (36, 48) | −330.71 | −394.09 | −368.00 |

For LEO objects, the SMA differences between two arcs of the same object are all less than 5 km, regardless of the time interval. However, the SMA difference is −27.5 km when the time interval between two arcs of different objects is less than 12 h. When the time interval increases, the SMA difference also increases. Again, the SMA differences with respect to the two arcs of different objects are significantly larger than those for two arcs of the same object.

Because of the relatively small difference between the estimated SMA and reference SMA when two arcs are from the same object, Equation (14) will very likely hold. On the other hand, Equation (14) will not most likely hold for two arcs from two different objects.

With the above validation, we now perform the two-arc associations for the 3179 arcs of simulated data that have successful IOD orbits. The true positive (TP) rate, which is the probability to correctly associate two arcs from the same object, is given in Table 5. The used association methods include the one by Wang et al. [13] and the proposed method in this paper.

**Table 5.** The TP rates of associating simulated GEO arcs with two different methods.

| Method | Interval ≤ 0.5 d | 0.5 d < Interval ≤ 1.5 d |
|---|---|---|
| Method in Wang et al. [13] | 85.66% | 63.89% |
| Proposed method | 93.10% | 73.57% |

It can be seen from Table 5 that the proposed method outperforms the method in Wang et al. by large margins in the TP rate [13]. On the errors of the determined SMA using two arcs from the same object, it is found that 84.54% of the SMA errors are less than 20 km, and 81.36% of them are less than 10 km with the proposed method. This demonstrates significant improvement over the IOD accuracy using single short arcs presented in Table 1.

In addition, the shorter the time interval between two arcs, the higher the TP rate is, because the two arcs can be more accurately modelled by a two-body orbit. When the time interval is less than half a day, the TP rate is 93% using the proposed method. It is understandable that the short re-surveillance interval is vital to the efficient autonomous cataloging, independent of the optical or radar data.

Table 6 shows the results using the proposed two-arc association method to process real data. The TP rates for GEO objects are 78.45% and 90.90% for Changchun GEO EA and SAO FocueGEO, respectively, when the time inverval is less than 12 h. They are 99.84% and 85.76%, respectively, when the time inverval is between 24 h and 36 h. For LEO objects, the TP rate is 100% when the same interval is between 24 h and 36 h, and it is 80.97% when the time interval is less than 12 h. These results demonstrate that the proposed method is effective in performing two-arc association for both GEO and LEO objects with orbit eccentricities close to zero.

**Table 6.** The TP rates of associating real ground-based arcs with the proposed method.

| Sensor | Interval ≤ 0.5 d | 0.5 d < Interval ≤ 1.5 d |
|---|---|---|
| Changchun GEO EA | 78.45% | 99.84% |
| SAO FocusGEO | 90.90% | 85.76% |
| Changchun LEO EA | 80.97% | 100.00% |

It is worth noting that the algorithm for two-arc association uses the virtual ranges that are computed using the single-arc IOD SMAs at the very beginning of the iteration. However, the single-arc IOD accuracy is a less important factor as long as it can cause the iterative two-arc orbit determination using Equation (11) to converge. From the data-processing experiments, an error of 200 km in the single-arc IOD SMA appears to be acceptable for starting the iteration.

*3.4. Object Cataloguing*

Operation of the two-arc associations has generated a number of two-arc pairs, and the two arcs forming a pair are believed to be from the same object. Following the procedure discussed in Section 2.3, when a GEO object has two two-arc pairs involving three arcs, for example Arc 1–Arc 2 pair and Arc 2–Arc 3 pair, the three arcs can be processed in the least-squares approach, with Equation (10) as the measurement equations. If the parameters in

Equation (11) are all less than the preset threshold, all three arcs are believed to be from the same object, and the resulting orbit elements will be more accurate than that from two arcs. In this way, the object is essentially catalogued. The procedure is applicable to more arcs, and thus the accuracy of the orbit elements can be improved, and autonomous cataloguing is achieved.

Figure 5 shows the accuracy improvement of the estimated SMA (a), eccentricity (b), inclination (c), and three-dimensional position (d, e and f) when the number of used arcs increases in the autonomous cataloguing of GEO objects with the observations observed by the simulated space-based sensor, the Changchun GEO EA, and the SAO FocusGEO, respectively. It is seen that when 4 or more arcs are processed, the errors of the three orbit elements are all close to zero, and the 3D (three-dimensional) position errors are stable at about 25–30 km.

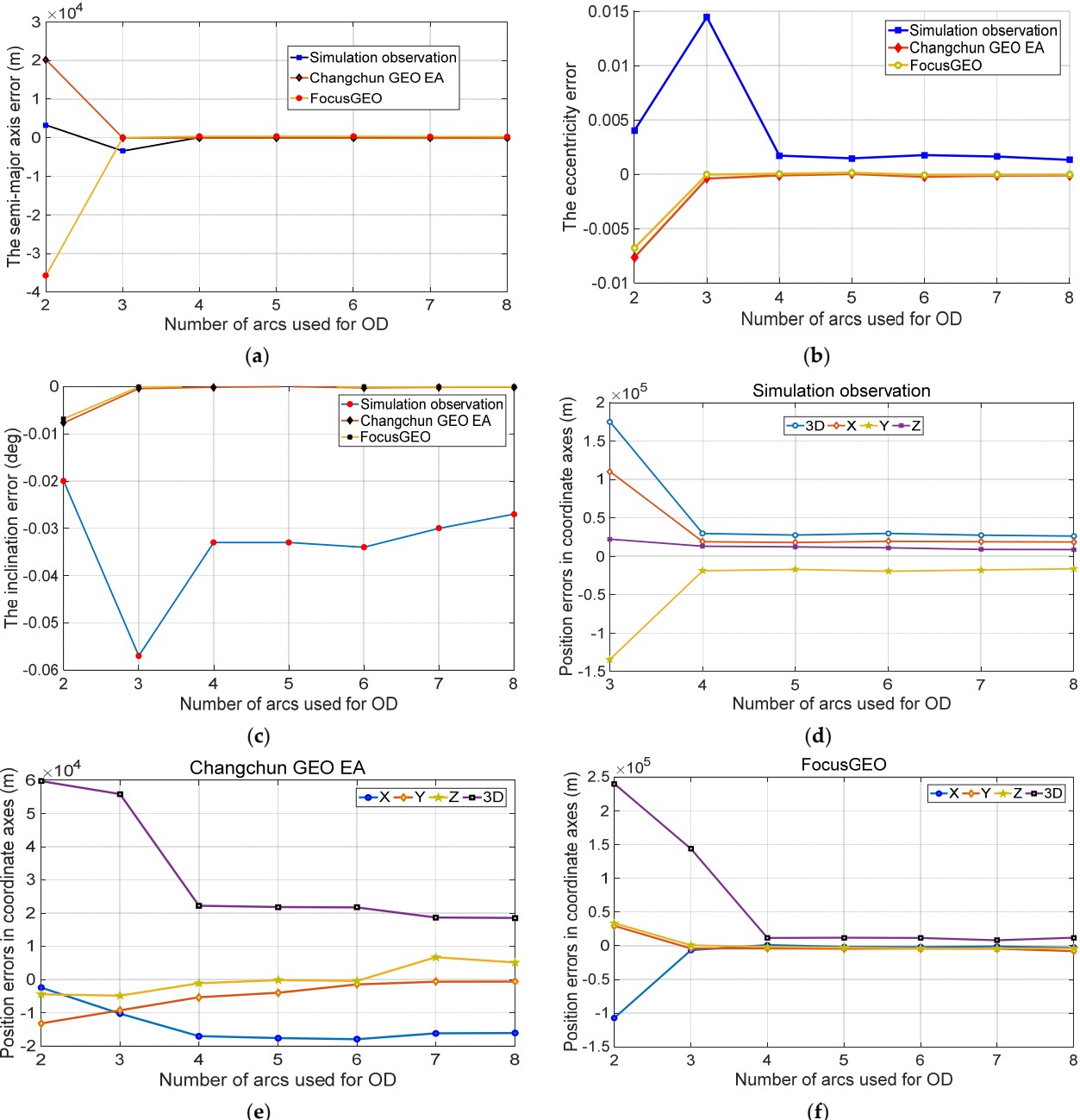

**Figure 5.** Errors of the semi-major axis (**a**), eccentricity (**b**), inclination (**c**), three-dimensional position with simulation observation (**d**), Changchun GEO EA (**e**), and the FocusGEO (**f**), respectively, after the orbit determination using multiple arcs.

Applying the procedure to both the simulated space-based angle data and real ground-based angle data, we have Figure 6 showing the overall cataloguing performance in terms of the TP rate and OD errors against the number of used arcs.

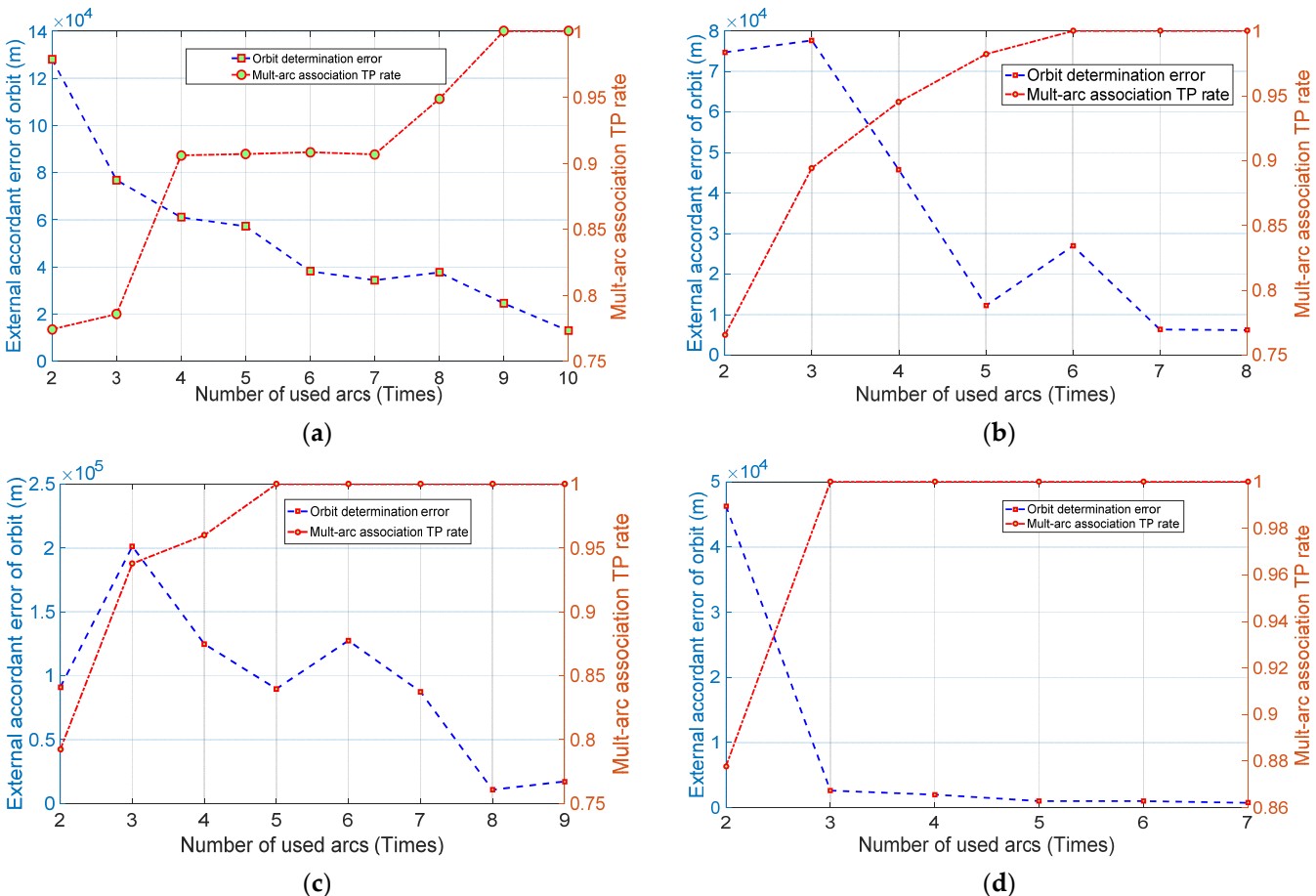

**Figure 6.** Overall cataloguing performance with different observations against the number of used arcs. (**a**) The space-based simulation data; (**b**) The real data of GEO objects from Changchun GEO EA; (**c**) The real data of GEO objects from FocusGEO; (**d**) The real data of LEO objects from Changchun LEO EA.

It can be seen from Figure 6 that with more arcs successfully used in the orbit determination, both the TP rate and 3D position accuracy are rapidly improved. When four or more arcs are used, the TP rates are all stably over 90%, meaning that autonomous and stable cataloguing has been achieved. In fact, when four arcs are used, the TP rates are all stable at more than 95% for ChangchunGEO EA, SAO FocusGEO EA, and ChangchunLEO EA. At the same time, the orbit determination accuracy is increased with the increasing number of observation arcs used. The accuracy is improved from 128 km with only two arcs to about 15 km with 10 arcs for the simulation data. In particular, when eight arcs are used, the errors are several kilometres for Changchun GEO EA. For Changchun LEO EA, the errors are stable at several kilometres when three or more arcs are used. Such accuracy is sufficient to perform rigorous least-squares orbit determination using multiple arcs.

Higher OD accuracy from multiple arcs can be expected if more perturbation forces, such as solar radiation pressure, are considered. However, it is beyond the scope of this paper.

## 4. Conclusions

Space-based surveillance of space objects appears to be an effective complement to regional ground-based surveillance, particularly for GEO objects. However, it is still a challenging task to autonomously associate short orbit arcs of angular data, and reliably

and accurately catalogue optically detected new objects. The cause for such difficulty is the lack of ranges between the optical sensor and object, and the short length of the orbital arcs, which makes the orbital geometry strength weak for the orbit determination. Therefore, a straightforward strategy is to "generate" range-related information and use it in the object orbit determination. For an object in the GEO orbit region, its orbit eccentricity is usually small, and the semi-major axis is approximately known within a few hundreds of kilometers. Assuming a value for the semi-major axis, when the position of the sensor and the unit vector from the senor to the object are known, the virtual range between the sensor and the object can be computed.

Based on the extensive use of the virtual ranges, this paper proposes a three-step approach to autonomously catalogue GEO objects from short angular orbit arcs. The three steps are aimed, respectively, at the short-arc initial orbit determination, the association of two arcs, and orbit determination using multiple arcs. In the first step, a multi-point optimal IOD method is developed to improve the convergence rate and the accuracy of the angles-only short arc IOD. The second step uses the Lambert equation to associate two arcs, and the decision on the correlation of two arcs is made based on the magnitude of the deviation rates of the residuals of the angle data. The third step is to determine the orbit using three or more arcs with the least-squares method.

Experiments with the simulated space-based angle data of GEO objects show that when the arc length is only 3 min and the errors of the right ascension and declination observations are 10″, a very high IOD success rate of 98.25% is achieved, the true positive rate of associating two arcs within 0.5 days of each other is about 93%, and the orbit determination accuracy with 10 arcs is stably at about 15 km. This provides practically usable techniques to catalogue new optically detected GEO objects.

The three-step approach is also tested with real ground-based angle data for both the GEO and LEO objects in near-circular orbits. The IOD success rates are all above 95%. For the critical TP rate of two-arc association, it is over 78% for GEO objects, and 80% for LEO objects. These results demonstrate that the proposed method is effective for processing real data.

It should be pointed out that, because of the assumption regarding orbit circularity, the proposed method will be not applicable to eccentric orbits, such as Molniya orbits.

**Author Contributions:** J.H. and J.S. worked on the conceptualization; X.L. and G.Z. worked on the methodology and software; Z.L. provided the Changchun EA observations; H.L. provided the SAO FocusGEO observations; L.L. worked on the validation test of the method and software, J.H. and X.L. processed the observations and wrote the paper; funding acquisition, J.S. All authors have read and agreed to the published version of the manuscript.

**Funding:** This research was partly funded by the National Natural Science Foundation of China, grant number 41874035.

**Informed Consent Statement:** Not applicable.

**Conflicts of Interest:** The authors declare no conflict of interest.

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
