# Peer review of "Short-Arc Association and Orbit Determination for New GEO Objects with Space-Based Optical Surveillance"

_aerospace, doi:10.3390/aerospace8100298_

Round 1

Reviewer 1 Report

This is a very timely and interesting investigation into the mounting problem of keeping up-to-date, accurate catalogues of Earth-orbiting artificial satellites. The authors have tackled the difficult problem of monitoring objects in GEO, where from sensors in LEO or on the ground, the use of angles-only observations leads to problems in orbit determination. The authors have devised a novel system of, with an assumption about the likely zero eccentricity and standard GEO semi-major axis, determining some short-arc parameters in a scheme to test whether several such arcs relate to the same object. That done, the combined short arcs together are used to produce a catalogue entry of sufficient accuracy to be of GEO-monitoring value. The work is so far based on simulations, and the authors are encouraged to use real observations, initially from ground-based sensors, to test their potentially very powerful process. Their final assertion that the method may be useful if also applied to LEO objects will require further research, because the basic initial assumptions of zero eccentricity and estimated SMA will of course have to be revised.

A few language improvements have been noted on the manuscript. 

Author Response

Dear Editors and Reviewers,

We would like to thank you for giving us the opportunity to revise our paper and thank you for your careful reviewing and valuable comments, which help us, improve the quality of our paper. Following the reviewers’ comments, we have made modifications to the paper, which hopefully has met the acceptance standards of Aerospace. In the revised manuscript, we have addressed, point by point, all the comments from the reviewers to eliminate our technical and structural deficiencies. We look forward to a positive decision on the paper acceptance.

For the convenience of the reviewers to read, all the revised or newly added parts in this manuscript are marked in red.

Reviewer 2 Report

This manuscript outlines methods to catalog GEO objects from short angular orbit arcs based on the virtual ranges between the sensor and the object. The manuscript is interesting but needs to address several major concerns.

What are the orbit prediction errors for the proposed algorithm for a certain period of time? Explain in detail with relevant plots.

Authors have shown "three-dimensional" position error in Figure 4; but this is not very clear; I would suggest authors provide error plots for different axes as well.

Also please elaborate on the trajectory error (position and velocity) prediction using the proposed method for different axes.  The authors have only provided a 3-dimensional position error, this is not adequate to estimate the performance of orbit determination.

The authors have provided details of orbital determination and error analysis mostly on the semi-major axis; what about other orbital parameters? 

What are the effects of different noises such as white Gaussian, colored noises on the accuracy of the proposed method? Please elaborate with figures.

Provide the step-by-step algorithm detailing the implementation of the proposed method. 

What is the justification to assume a 10" error in right ascension and declination in the proposed method?

How the proposed method can be adapted to catalog objects in LEO? Please elaborate with an example.

I would recommend authors apply the proposed method to actual data if they are available for validation.

What are the error bars in Fig. 4 and Fig. 5?

Author Response

(The authors gave the same response as above.)

Reviewer 3 Report

The manuscript is devoted to an important problem of initial orbit determination of objects at geosynchronous Earth orbit (GEO) from space-based angular optical surveillance. The number of GEO objects is increasing and, therefore, surveillance and determination of orbits of these objects is important to propagate their positions, prevent their collisions, and to find “windows” for new GEO satellites. Authors performed a detailed review of existing methods of initial orbit determination, in particular, in application to GEO objects and proposed a new three-step approach of the initial orbit determination of these objects from angular observations based on the use of virtual ranges between an optical sensor and GEO object. Using simulated observations authors demonstrate better performance of their approach, as compared to a few previously published methods. The manuscript is good structured and generally rather good written. The authors should take into consideration the following comments, before a decision can be made on the acceptance of the manuscript for publication.

General comments

1. The approach proposed by authors is based on an assumption that GEO objects have circular orbits. In fact, most of them have orbits will small eccentricity, i.e. almost circular. So, it seems the proposed algorithm can’t be applied for objects with Molnija-like elliptical orbits having apogee at the altitude of about geosynchronous orbit and perigee at lower altitude, right? I think, one should discuss (mention) this in the manuscript. I think, it is important to provide in the introduction a value of the orbit determination accuracy for GEO satellites required for reliable initial orbit determination and identification of satellites tracked in a few arcs.

2. It is not clear how the threshold in Eq. (11) is defined. It should be explained. Some extensive simulations to set the thresholds are meant in lines 428-429. I think, some details on these simulations should be given.

3. I wonder, how competitive is the approach being proposed in this manuscript and providing an accuracy of about 15 km from about 10 arcs of about 3 minute length each, when some published approaches and methods provide better accuracy, e.g. 0.05 to 1.80 km in the radial direction and smaller values in two other directions (Scott and Ellery, 2015), 0.01-0.10 km (Tombasco and Axelrad, 2011)?

Specific comments

These comments refer to the line numbers of the original manuscript given in the form Lxxx.

L18: I would add a word “observations” after “angles-only”.

L26: “at an accuracy of about 15 km”. Is it three-dimensional (3D) accuracy or accuracy along the orbit? Please, specify.

L29-30: I would add “geosynchronous orbit” in the list of keywords.

L34: the words “Space powers” are not clear.

L38: The list of applications of GEO satellites should be longer, including some scientific applications. Some references on GEO satellite applications should be added.

L41: one important application of cataloging of GEO objects is also to prevent possible collisions of GEO objects.

L42-43: please provide some references for ground-based optical telescopes performing surveillance of GEO objects.

L47-49: I wonder, if satellite(s) for optical surveillance of GEO object already exist(s)? Are they routinely used or are they just in planning and preparation?

L51: since the term “arc association” is widely used in the manuscript, it should be explained at its first used.

L54 and many other places in the manuscript: I think, one should use the expression “angle data” instead of “angles data”.

L63: duration of a short and very short arcs should be specified.

L63-64: it is not clear, what is meant under the word “their” in the statement “IOD results are the very base of their association”. Do you mean “IOD results are the very base of arc association” Please, write it more clear.

L67: “since” should read “due to”.

L78: “Since” should read “due to”.

L79: “in limited precision” should read “of limited precision”.

L88: “a useful tools” should read “a useful tool”.

L96: the sentence starting with the words “While Sang et al. Proposed” should be a continuation of the previous sentence.

L105: the words “Stoker et al. analyzes” should read “Stoker et al. analyze”.

L107: the words “the amount of observation arc” are not clear. Do you mean “the amount of observation arcs”?

L109: no “of” is necessary inside “determination of are”.

L114: it seems, the words “the IOD convergence rate and accuracy of the IOD” should read “high IOD convergence rate and accuracy of the IOD”.

L119-120: the words “the estimated orbit using” should read “the orbit estimated using”.

L123: the word “while” is not necessary before “which is usually unavailable”.

L128: the words “circularity of most GEO objects” should read “circularity of orbits of most GEO objects”, since you mean circularity of orbits, not of objects.

Figure 1: nothing is written in the figure about virtual ranges that are an important part of the proposed approach.

L153: “period25” should read “period”.

L156: one normally writes “con-verge” as “converge”.

L161: the words “a property of GEO orbits to determine the IOD elements” should be reformulated, since the orbits do not determine the elements by themselves.

L172-173: J2 is the second order term of the Earth gravitational expansion.

L173: are the r1 and r2 vectors the geocentric position vectors?

L203: the first term on the right hand side of the formula for a_i should be a_1, not a_i.

L239: the text starting with “It is then easy to compute the RMS” should begin at a new line. The abbreviation RMS should be explained.

L246: a space between C_0 and “and” should be given.

L251: the words “Eq(5) below” should be placed in parentheses.

L268: a space between the word “theory” and the reference number “[27]” should be given. The same refers to similar cases in lines 288, 289, 292, 296, where a space before the reference number is missing.

L305-306: the sentence starting with “Before proceeding to the details” is not complete.

L313-315: formulae in these lines should be numerated and provided with a reference.

L322-323: the following statement should be explained: “These values demonstrate that the GEO orbit over a 3 day time span could still be described by a two-body orbit in the two-arc association”.

L377: the words “the Earth gravity of order/degree 5” should read “the Earth gravity of order/degree up to 5”.

L421: “of observed GEO arc” should read “of an observed GEO arc”. The words “the term of “d”” should better read “the term “d””.

L429-430: an empty line should be added between two these lines.

L435: “The IOD is claimed a success if the error of the estimated SMA is less than 1000km.” Who defined this large value? It is really large!

L456: “much larger errors”. Do you mean “much larger differences”?

L481: “two arcs forming a pair is believed” should read “two arcs forming a pair are believed”.

Figure 4: a space before “(km)” should be added in the text of the Y axis of both figures.

Figure 5: a space should be added before “(Times)” in the text of the X axis. For consistency, the same units should be used in the Y axes of Figures 4 and 5: either km or meter. Three different words are used in Figures 4 and 5: error, accuracy and precision. Do the author make difference between them? Is the 3D accuracy is meant under “accuracy” in the Y axis of Figure 5?

L505 and L534: do you mean 3D accuracy? If so, it should be specified.

L507-508: please, provide a reference to support this statement.

L510: “appears an effective complement” should read “appears to be an effective complement”.

L515: not sure, that the word “strength” is necessary in this sentence.

L550: the words “EUROPEAN SPACE AGENCY-PUBLICATIONS-ESA SP.” should read “European Space Agency Publications - ESA SP.”.

L581: “German” should read “Germany”.

L582-583: I couldn’t find the reference 18 (Stoker et al., 2020). A doi number or web link should be provided.

L590-591: the journal name of the reference 22 should be written completely, in consistency with the references 2, 7, 8, 9, 13.

References 23, 25, 26 and 35 (in Chinese) do not seem to be easily accessible. Can you either provide web links to them or replace these references by other references in English that are accessible?

References given in the review:

Scott, R. L. and Ellery, A. An approach to ground based space surveillance of geostationary on-orbit servicing operations, Acta Astronautica, Volume 112, July–August 2015, Pages 56-68, https://doi.org/10.1016/j.actaastro.2015.03.010.

Tombasco, J., Axelrad, P. A Study of the Achievable Geosynchronous Angles-Only Orbit Estimation Accuracy. J of Astronaut Sci., 58, 275–290 (2011). https://doi.org/10.1007/BF03321169.

Author Response

(The authors gave the same response as above.)

Round 2

Reviewer 2 Report

Thanks authors for addressing my comments; I am happy with that. The revised manuscript has been improved significantly. 

Author Response

Response to Reviewer #2

Dear Reviewer,

 Thank you for your careful reviewing and valuable comments, which help us, improve the quality of our paper significantly.

Reviewer 3 Report

The authors took into consideration my comments to the original version of the manuscript. They also added some results on using real observations of GEO and LEO objects to support the conclusions made using simulated observations. The manuscript is generally improved and is, from my point of view, on a good way to be accepted for publication. There are, however, some minor issues to be fixed before this decision can be made. In particular, English should be improved. I have made below some suggestions. Also, a few not completely clear things should be clarified in the manuscript. They are given below.

Specific comments

These comments refer to the line numbers of the revised manuscript given in the form Lxxx.

L29-30: it is better reformulate “at an accuracy of about 15 km in 3-dimensional” as “at 3-dimensional accuracy of about 15 km”.

L41-44: the sentence starting with the words “Sensors on a Geosynchronous Earth Orbit...” should be better split into two sentences, since, in fact, GEO satellites, not sensors are used “in communications, reconnaissance, weather predication, defense applications, scientific applications, etc.”

L73: “short-arc or very short-arc (VSA) angles (less than 1% of orbital period)”. If I correctly understand, the words “less than 1% of orbital period” refer to a very short-arc angle arc. Please, provide a duration of a short arc to distinguish between the short and very short arc duration.

L223: “the function values at the lower and upper boundaries of a SMA sub-range is” should read “the function values at the lower and upper boundaries of a SMA sub-range are”, i.e. the word “are” should be used instead of “is”, since the word “values” is plural.

L277: a comma is missing after the words “for example”.

L283: R and V in Eq. (6) should be explained. They are not presently explained.

L315: the word “difference” should be plural: “differences”.

L316: normal vectors to what? It should be specified.

L359: an explanation of the second order norm of a vector should appear already in L202, when it is used for the first time in the manuscript.

L375: the words “the correlation of the two arcs are judged” should read “the correlation of the two arcs is judged”, since the word “ correlation” is single.

L382: the words “the eccentricity of GEO object is nearly zero” should, probably, read “the eccentricity of GEO is nearly zero”, since one can specify eccentricity of an orbit, but not of an object.

L419: it is better to use “final result” instead of “end result”.

L439: the words “orbit elements is denoted” should read “orbit elements are denoted”, since the word “elements” is plural.

L442: the text “can be declared from the same object” should be better written as “can be declared to be from the same object”.

L453-473: I would suggest to put the text with the description of the real data used in the manuscript just after the L521, where the results on using these data are presented. Presently the description of the real data and their results are interrupted by the results on simulated data.

L457: a reference on the Changchun GEO EA telescopes would be very useful at this place.

L457: the abbreviation FOV (field-of-view) is explained in the manuscript 4 times: in L42, where it is used the first time, and in L457, L468, L481. One should either explain it once, or use use the complete words “field of view” in all cases.

L511: the words “the proposed one in this paper” should better read “the one proposed in this paper”.

L595: the text “with orbit eccentricities close to zero” should be added at the end of the sentence “These results demonstrate that the proposed method is effective to performing two-arc association for both GEO and LEO objects.”

L599: the wording in the text “that are computed the single-arc IOD SMA at the very begin of the iteration” should be improved. It is not clear. The word “begin” should read “beginning”.

L600-601: the text “it can make the iterative two-arc orbit determination using Eq (11) converge” should read “it can make the iterative two-arc orbit determination using Eq (11) to converge”, i.e. the words “to converge” should be used instead of “converge”.

L610-611: “are believed from the same object” should read “are believed to be from the same object”.

Figure 6: the text in the Y axis of the upper right panel (“External accordant error of orbit (m)”) should be written using the same font size as those in three other panels of this figure. Presently, it is a bit larger.

L640: “against number” should read “against the number”.

L649, L650: a typo in “serveral” should be fixed: “several”.

L661-662: what do you mean under “optical arcs”? Do you mean “orbital arcs”?

L678: “dat” should read “data”.

L772 and L788: please check, if Ref. 37 and Ref. 46 are the same and one of them can be erased.

Author Response

Dear Reviewer,

We would like to thank you for giving us the opportunity to revise our paper and thank you for your careful reviewing and valuable comments, which help us, improve the quality of our paper. Following the reviewers’ comments, we have made modifications to the paper, which hopefully has met the acceptance standards of Aerospace. In the revised manuscript, we have addressed, point by point, all the comments from the reviewers to eliminate our technical and structural deficiencies. We look forward to a positive decision on the paper acceptance.

For the convenience of the reviewers to read, all the revised or newly added parts in this manuscript are marked in red.
